# Effect of Cigarette Smoking on Clinical and Molecular Endpoints in COPD Patients

**DOI:** 10.3390/ijms25115834

**Published:** 2024-05-27

**Authors:** Patrizia Russo, Francesca Milani, Antonio De Iure, Stefania Proietti, Dolores Limongi, Carla Prezioso, Paola Checconi, Vincenzo Zagà, Federica Novazzi, Fabrizio Maggi, Guido Antonelli, Stefano Bonassi

**Affiliations:** 1Department of Human Sciences and Promotion of the Quality of Life, San Raffaele University, Via di Val Cannuta 247, 00166 Rome, Italy; patrizia.russo@uniroma5.it (P.R.); francesca.milani@uniroma5.it (F.M.); stefy7677@gmail.com (S.P.); dolores.limongi@uniroma5.it (D.L.); carla.prezioso@uniroma5.it (C.P.); paola.checconi@uniroma5.it (P.C.); stefano.bonassi@sanraffaele.it (S.B.); 2Clinical and Molecular Epidemiology, Istituto di Ricovero e Cura a Carattere Scientifico—IRCCS San Raffaele Roma, Via di Val Cannuta 247, 00166 Rome, Italy; 3Experimental Neurophisiology Lab, Istituto di Ricovero e Cura a Carattere Scientifico—IRCCS San Raffaele Roma, Via di Val Cannuta 247, 00166 Rome, Italy; 4Laboratory of Microbiology, Istituto di Ricovero e Cura a Carattere Scientifico—IRCCS San Raffaele Roma, Via di Val Cannuta 247, 00166 Rome, Italy; 5Italian Society of Tabaccology (SITAB), Via G. Scalia 39, 00136 Rome, Italy; vincenzo.zaga@icloud.com; 6Department of Medicine and Surgery, University of Insubria, Via Ravasi 2, 21100 Varese, Italy; federica.novazzi@uninsubria.it; 7Istituto Nazionale Malattie Infettive Lazzaro Spallanzani, Via Portuense 292, 00149 Rome, Italy; fabrizio.maggi.63@gmail.com; 8Virology Laboratory, Department of Molecular Medicine, Sapienza University, Viale Porta Tiburtina 28, 00185 Rome, Italy; guido.antonelli@uniroma1.it; 9Microbiology and Virology Unit, Sapienza University Hospital Policlinico Umberto I, Viale del Policlinico 155, 00161 Rome, Italy

**Keywords:** cigarette smoking, COPD, DNA damage, mortality, virus, inflammation

## Abstract

Cigarette smoking is a primary contributor to mortality risks and is associated with various diseases. Among these, COPD represents a significant contributor to global mortality and disability. The objective of this study is to investigate the effect of smoking on a selected battery of variables, with an emphasis on DNA damage. A total of 87 elderly patients diagnosed with COPD, divided into three groups based on their smoking history (current, former, never-smokers), were evaluated using a cross-sectional approach. Clinical features including mortality and inflammatory/oxidative parameters (Lymphocytes/Monocytes, Neutrophils/Lymphocytes, Platelets/Lymphocytes ratio), SII, MDA, 8-Oxo-dG, and IL6 (ELISA assay), as well as DNA damage (comet assay), were investigated. Virus infection, i.e., influenza A virus subtype H1N1, JC polyomavirus (JCPyV), BK polyomavirus (BKPyV), and Torquetenovirus (TTV), was also tested. Current smokers exhibit higher levels of comorbidity (CIRS; *p* < 0.001), Platelets/Lymphocytes ratio (*p* < 0.001), systemic immune inflammation (*p* < 0.05), and DNA damage (*p* < 0.001). Former smokers also showed higher values for parameters associated with oxidative damage and showed a much lower probability of surviving over 5 years compared to never- and current smokers (*p* < 0.0017). This study showed a clear interaction between events which are relevant to the oxidative pathway and cigarette smoking. A category of particular interest is represented by former smokers, especially for lower survival, possibly due to the presence of more health problems. Our findings raise also the attention to other parameters which are significantly affected by smoking and are useful to monitor COPD patients starting a program of pulmonary rehabilitation (DNA damage, inflammation parameters, and selected viral infections).

## 1. Introduction

Although epidemiological studies identified cigarette smoking as a primary contributor to mortality risks more than 50 years ago, various forms of smoked tobacco products, such as manufactured and hand-rolled cigarettes, pipes, cigars, waterpipes, bidis, kreteks, and other locally consumed smoked tobacco products, continue to be prevalent in numerous analyses from the global burden of disease study 2019 [1].

Cigarette smoking has been associated with the onset of cardiovascular diseases (CVD), chronic obstructive pulmonary disease (COPD), hypertension, cancer, and numerous chronic systemic diseases characterized by inflammatory processes. This prevalence has resulted in significant mortality and disability directly linked to tobacco use, making smoking the predominant cause of disease burden in many nations. Worldwide, an estimated 1.18 (95% CI: 0.94 to 1.47) billion individuals were habitual tobacco users in 2020, leading to 7.0 (95% CI: 2.0 to 11.2) million fatalities, representing roughly one-seventh of all deaths recorded during that period [1], including atherosclerosis, Crohn’s disease, rheumatoid arthritis, psoriasis, Graves’ ophthalmopathy, and type 2 diabetes [2,3]. The mechanisms by which cigarette smoking initiates and worsens these conditions are complex, interconnected, and not yet fully understood. Tobacco smoke comprises a complex and dynamic chemical blend containing 7357 compounds from various classes, which are either bound to aerosol particles or present in the gas phase. Furthermore, smoking habits (such as puff volume, number of puffs per cigarette, and percentage of blocked ventilation holes) significantly influence the levels of toxic, carcinogenic, and addictive substances delivered to the smoker in cigarette smoke. Smokers’ puffing patterns vary widely from individual to individual. Despite uncertainties regarding whether specific constituents of cigarette smoke are accountable for particular adverse health outcomes, there is broad scientific consensus that many of the primary chemical classes in the combustion emissions of burned tobacco are toxic and carcinogenic [4]. Numerous investigations have delved into the respiratory consequences of smoking, beginning with the groundbreaking research of Fletcher and Peto, published in 1977, which laid the foundation for our current comprehension of the association between smoking and diminished lung function, as well as the critical significance of smoking cessation in clinical practice [5].

COPD represents a significant contributor to global mortality and disability. It presents as a multifaceted and diverse lung condition characterized by persistent airflow obstruction and respiratory symptoms. Typical symptoms of COPD are dyspnea, fatigue, and reduced quality of life [6]. Tobacco smoking accounts for over 70% of COPD cases in high-income countries. In low- and middle-income countries, tobacco smoking accounts for 30–40% of COPD cases, and household air pollution is a major risk factor [7]. As of 2019, COPD stood as the third leading cause of death [8]. The development of COPD stems from interactions between genetic (G) and environmental (E) factors across an individual’s lifespan (T) (referred to as GETomics), which can lead to lung damage or disrupt their normal developmental and aging processes [9].

Tobacco smoking remains a key, preventable environmental risk factor for COPD [10], although not all smokers develop the condition. Moreover, large-scale studies have indicated that 22–51% of individuals with COPD have never smoked (defined as <100 cigarettes in a lifetime); conversely, estimates of COPD prevalence in never-smokers range from 4% to 16% [11]. Never-smokers with COPD typically exhibit milder chronic respiratory symptoms and airflow limitation compared to smokers with COPD, but still face a poor prognosis with an increased risk of exacerbations [11]. Thus, smoking exposure is not the only risk factor for low lung function through life.

It has been documented that dysanapsis, the disproportionate scaling of airway dimensions to lung volume, is significantly and independently linked to both the prevalence and incidence of COPD among older adults [12]. This finding may provide insight into why only a minority of individuals with a history of heavy smoking develop COPD and why up to 30% of COPD cases occur in individuals who have never smoked.

Tobacco smoking cessation represents the most effective intervention in halting the progression of COPD, thereby reducing morbidity and improving patient survival rates [11].

The mechanisms through which cigarette smoking initiates and exacerbates COPD are intricate, interconnected, and still not fully understood. A recent review suggests that inflammasomes (leucine-rich repeat (LRR)-containing proteins (NLR) family members such as NLRP3, NLRP6, NLRP12, and interferon-inducible protein (AIM2)) may contribute to the development of various diseases associated with cigarette smoke exposure, including COPD. Specifically, NF-κB activation through a MyD88-dependent pathway, the production of reactive oxygen species (ROS), endoplasmic reticulum stress, mitochondrial dysfunction, and calcium influx are potentially involved in the activation of inflammasomes induced by cigarette smoke [13]. Oxidative stress is increased in COPD patients, even in former smokers and never-smokers, and is further increased during acute exacerbations. Oxidative stress directly damages DNA [14]. It has been found that the number of DNA double-strand breaks (DSBs) in alveolar type I and II cells and endothelial cells was higher in COPD patients than in asymptomatic smokers and nonsmokers [14]. This data supports the suggestion that DSBs, at least partially caused by oxidative stress, seem to contribute to the pathogenesis of COPD by inducing apoptosis, cell senescence, and pro-inflammatory responses.

A recent review [15] summarizing the extent of DNA damage in highly prevalent diseases reported a higher occurrence of DNA strand breaks in patients with COPD compared to control groups (2.25-fold, 95% CI: 1.28-, 3.23-fold, n = 6). However, the meta-analysis showed higher levels of DNA strand breaks in the group of COPD patients (SMD = 1.29, 95% CI: 0.69, 1.90). The impact of current smoking habits was observed in a study where smoking-related COPD patients had higher levels of DNA strand breaks than non-smoking biomass-related COPD patients and non-smoking controls without COPD [16]. Other studies have shown an over-representation of current smokers in the COPD patient group and reported higher levels of DNA strand breaks in COPD patients compared to controls [17,18,19]. Adjustment for current smoking and other confounders decreased the odds ratio for elevated levels of DNA strand breaks with increasing severity of COPD, although the effect size was relatively small in adjusted analyses (e.g., odds ratio = 1.008 for an increase in %DNA in tail per increment in COPD severity group) [20].

In addition to directly harming the pulmonary system, cigarette smoke is a well-known contributing factor to the onset and exacerbation of infectious diseases caused by both bacteria and viruses, such as SARS-CoV-2 [21] and influenza virus [22]. Thus, cigarette smoke and many of its constituents induce structural alterations in the respiratory tract [22]. These alterations include peribronchiolar inflammation and fibrosis, increased mucosal permeability, impairment of mucociliary clearance, changes in pathogen adherence, and disruption of the respiratory epithelium. These changes are believed to increase susceptibility to upper and lower respiratory tract infections, potentially exacerbating cigarette smoke-induced lung inflammation.

On average, smokers exhibit an elevated peripheral white blood cell count, approximately 30% higher than that of nonsmokers, with increases observed in granulocytes, lymphocytes, and monocytes; the differential leukocyte count, however, shows no significant change [23]. Moreover, it has been shown that smokers with symptoms of chronic bronchitis and chronic airflow limitation have an increased number of CD8+ T-lymphocytes compared to asymptomatic smokers with normal lung function [24]. Traditionally, the primary function of CD8+ T-lymphocytes has been considered the rapid resolution of acute viral infections [25], which are common in patients with COPD. As previously suggested [26], it is possible that an excessive recruitment of CD8+ T-lymphocytes may occur in response to repeated viral infections in some smokers, and this excessive response may play a crucial role in the development of pulmonary damage in these subjects [24].

It is evident that exacerbations of COPD are often linked to viral and/or bacterial infections [27,28]. While the exact percentages vary among individual studies, it is generally observed that around 20–30% of patients experiencing acute exacerbations of COPD have a detectable bacterial infection, 20–50% have a viral infection, and approximately 25% have a bacterial–viral co-infection [29]. Rhinovirus is the primary viral pathogen identified in COPD exacerbations [30]. While the detection of viruses during acute COPD exacerbations confirms an association, it difficult to establish a causal role in exacerbation pathogenesis.

In this study, the effect of smoking on a selected battery of variables, with an emphasis on DNA damage, was evaluated in a population of COPD patients using a cross-sectional approach. This was supplemented by a five-year prospective follow-up concerning mortality. Moreover, the study explores the relationships between smoking status and virus presence. Specifically, we selected influenza A virus subtype H1N1 and human polyomavirus (HPyV) JC (JCPyV) and BK (BKPyV), considering that all patients were under corticosteroid therapy. It is well known that HPyVs, often latent in healthy individuals, may be reactivated under immunosuppressive conditions [31]. Additionally, we investigate Torquetenovirus (TTV), not only for its prototype role in the Anelloviridae family [32] but also because we showed that a TTV load ≥4 log copies/mL is associated with an increased risk of mortality in the frail population [32].

## 2. Results

### 2.1. Demographic and Clinical Characteristics

A total of 87 elderly patients (mean age ± SD: 72.38 ± 8.7 years, 45 females) were enrolled upon hospital admission after providing informed consent. The severity of COPD, categorized based on GOLD (Global Initiative for Chronic Obstructive Lung Disease) stages based on the forced expiratory volume (FEV-1), indicates that all patients fall between stage 3 and 4 (30% ≤ FEV-1 < 50%, FEV-1 < 30%, respectively; Table 1). Distribution by smoking habit, confirmed by cotinine assay, shows 32 current smokers, 45 former smokers, and 10 non-smokers. No difference among the three groups was found in the distribution of patients by any of the baseline variables considered including the ability to cover a distance (meter) over a time of 6 min (6 Minute Walk Test: 6 MWT), need for supplemental oxygen nor in St. George’s Respiratory Questionnaire (SGRQ) or in Maugery Respiratory Failure (MRF26), measuring the impaired health in chronic airflow limitation (Table 1).

Looking at the score obtained with the cumulative illness rating scale (CIRS), current smokers show a value of 3.58 ± 1.02, higher than former (2.65 ± 1.39) or never-smokers (1.71 ± 0.76).

There were no significant differences in Quality of Life (QoL) (Activities of Daily Living [ADL], Instrumental Activities of Daily Living [IADLs], [SGRQ]) values among the three patient groups, and they did not exhibit cognitive (Mini-Mental State Examination [MMSE]), anxious (Beck Anxiety Inventory [BAI]), or depressive (Beck’s Depression Inventory [BDI]) issues. Regarding lifestyle, no differences were found in terms of fruit and vegetable consumption among the three groups, nor in terms of physical activity, which was extremely limited in all subjects (not exceeding 30 min 2/3 times a week).

The table reports all parameters evaluated in relation to smoking habits in COPD patients. Variables include demographic, clinical, and biochemical data in addition to DNA damage and virus infection. The table highlights differences and trends associated with smoking status. Legend: malondialdehyde (MDA), 8-Oxo-2′-deoxyguanosine (8-Oxo-dG) or interleukine-6 (IL6), influenza A virus subtype H1N1, JC virus (JCPyV), BK polyomavirus (BKPyV), Toqueteno virus (TTV), and NS: not statistically significant.

### 2.2. Oxidative and Inflammatory Parameters

No differences between groups were observed in terms of oxidative parameters, as shown in Table 1. Looking at the inflammatory parameters, the ratio Pt/Lym (platelets over lymphocytes) was higher in current than in former or never-smokers (Table 1); also, the Systemic Immune-inflammation Index (SII) was higher in current than in former and never-smokers (Table 1).

### 2.3. DNA Damage

DNA damage evaluated by comet assay was quantified in terms of tail intensity (%). Table 1 shows that the tail intensity (19.47 ± 7.37) was elevated in all COPD patients. Thus, in a previous study analyzing a dataset of 8293 subjects [33], we reported a tail intensity value of 7.4 ± 8.8, with a value of 10.5 ± 11.2 for individuals over 60 years (1329 subjects). Tail intensity differed between current and former smokers compared to never-smokers with current smokers having the higher amount of DNA damage than former or never-smokers, respectively (Table 1). Tail intensity increased after 3 weeks of pulmonary rehabilitation in all groups of patients.

Since some patients were receiving oxygen therapy, we conducted an analysis to assess whether oxygen had an impact on DNA, and we found that there was no significant difference (Table 1).

### 2.4. Identification of Virus in COPD Patients

The identification of common viral infections, such as influenza A (H1N1), JCPyV, BKPyV, and the TTV viral infections, was performed in all patients. Table 1 shows the presence of infection in the study patients stratified by smoking habit. Influenza A virus (H1N1) was present only in one patient, and only the presence of JCPyV was statistically associated. However, no difference was associated between never-, current or former smokers. The presence of any virus did not influence the amount of DNA damage in all groups of patients.

### 2.5. Survival

A difference in all-cause mortality between current, former, and never-smokers has been observed after a follow-up of 5 years (*p* < 0.017), where former smokers have a much lower probability of survival than never- and current smokers (Figure 1).

## 3. Discussion

In this study, we assessed a total of 87 elderly patients diagnosed with COPD, divided into three groups depending upon their smoking history, i.e., current smokers, former smokers, and never-smokers. The presence of cotinine in urine served as an objective measure of smoking, increasing the overall prevalence of smokers from the self-reported rate of 35.5% to a more realistic 40.45%.

Since all enrolled patients were in severe GOLD stages 3 and 4, no clear differences were observed among the three groups for most clinical or functional parameters, including severity, 6 MWT, and BMI status. Accordingly, no disparities were found in terms of oxygen supplementation, quality of life, cognitive, anxious, or depressive scores. Regarding lifestyle, no differences by smoking status were found concerning fruit and vegetable consumption, or by physical activity, which was however limited in all subjects to no more than 2 h per week. 

A significant difference in comorbidity rate was observed among the three groups, in agreement with recent findings indicating that the number of comorbidities among patients with COPD influences the risk of moderate-to-severe exacerbations and correlates with the severity of respiratory symptoms and lung function [34]. On average, a typical COPD patient experiences five additional conditions, which are often unidentified or misdiagnosed and untreated or improperly managed [35]. In our COPD patients, 30% of never-smokers, 21.9% of current smokers, and 40% of former smokers suffered from type 2 diabetes, while cardiovascular diseases (CVD) were present in 25%, 45%, and 40%, respectively.

The mechanisms underlying multimorbidity in COPD are complex and varied, with smoking potentially exacerbating these mechanisms. Inflammation has been proposed as one of the mechanisms involved, although randomized controlled trials assessing the effectiveness of specific systemic anti-inflammatory agents on COPD outcomes have produced negative results [36,37]. Furthermore, the extensive SUMMIT RCT carried out in COPD patients with significant cardiovascular risk factors, which employed the combination of inhaled corticosteroids and long-acting β2-adrenoceptor agonists (ICS–LABA) to manage airway inflammation, did not affect mortality or cardiovascular outcomes [38]. These results suggest that tackling inflammation alone is not enough to modify COPD progression. In our sample we observed a higher Pt/Lym ratio in current smokers (Table 1), a biomarker which was associated with systemic inflammation not only in rheumatologic diseases and cancer but also in COVID-19 and various respiratory diseases [39]. The systemic immune-inflammation index, a widely used biomarker of inflammation, was associated with overall survival, progression-free survival, and responsiveness to immunotherapy among cancer patients [39] and was higher in the group of current smokers (Table 1), indicating increased risk of cardiovascular diseases [39]. However, in the present study, we did not find any differences in oxidative markers such as MDA, 8-Oxo-dG, or IL6 by smoking status. The interpretation of these results should take into account that all patients were under treatment with known antioxidant drugs such as corticosteroids and N-Acetyl-L-cysteine [40,41,42].

Tobacco smoke is recognized for causing adducts on DNA, which precede DNA damage and chromosome mutations [43]. In our study, we assessed DNA damage using the comet assay, which revealed higher levels in COPD patients compared to the unaffected population and in smokers compared to never-smokers. Despite the presence of contrasting results in [44], which reported no association between cigarette smoking and DNA damage, our study revealed an increased DNA damage in current smokers compared to former and never-smokers. These data align with the findings of a meta-analysis conducted by Hoffmann et al. [45], suggesting higher levels of DNA damage in smokers. Clear variations between smokers and non-smokers were also observed in smaller-scale studies [46,47,48,49]. Oxygen supplementation, a well-known factor capable of increasing the level of DNA damage [50], did not impact the amount of DNA damage in the three subgroups investigated.

Different viruses, such as JCPyV, BKPyV, and TTV, are present in all patients, although only JCPyV was significantly associated with smoking status (Table 1). The influenza A virus (H1N1) was present only in one patient who is current smoker. Based on our best knowledge, the presence of JCPyV in COPD patients is observed for the first time, particularly among current smokers. Regarding the presence of TTV, the data align with studies conducted in stable kidney transplant recipients, indicating a negative association between TTV load and urinary cotinine concentration [51].

The survival analysis over five years by smoking status revealed that former smokers have a much lower probability of survival compared to current and never-smokers (Figure 1). This data is apparently surprising because current smokers have a higher rate of comorbidity. On the other hand, former smokers have a higher percentage of type 2 diabetes (40%) compared to current (21.9%) and never-smokers (30%), and this could negatively influence the probability of survival. This result could also be attributed to the possibility that former smokers might have quit smoking due to health issues. Consequently, the overall health status of former smokers may be poorer compared to that of current smokers. A recent paper examining a total of 49,826 patients ≥40 years of age, with a hospital diagnosis of COPD in 2008–2017, reported during the 12-month follow-up period a 10% mortality, with a double proportion of former smokers as compared to current smokers [52]. A recent work shows that whereas current smokers has an increased inflammatory response following bacterial stimulation, which is quickly lost upon smoking cessation, the smoking effects on T cell responses persist years after individuals quit smoking, highlighting a mechanism for the persistent effects in the adaptive response [53]. The authors of the above article hypothesized that the persistent effect of smoking on adaptive immune responses is associated with DNA methylation at signal trans-activators and metabolism regulators.

COPD, as reported already [54], is a complex and heterogeneous disorder. “Complex” refers to the condition encompassing multiple elements with nonlinear relationships between them, such as FEV1, which is pivotal for diagnosis but may not always correlate directly with other factors. On the other hand, “heterogeneous” indicates that not all of these elements are present in every patient, or even in a particular patient, at all times (dynamic). Therefore, in such a heterogeneous disease, fatalities will occur due to diverse mechanisms at different stages of severity. Due to the complexity of COPD, these events occur both early and throughout the subject’s lifetime.

A description of most remarkable events occurring during the aetiopathogenesis of COPD and that may be of interest for the oxidative pathways are described in Figure 2 and Figure 3 commented on. Each individual event is in accordance with the most recent literature as follows: (a) Aging: the gradual decline attributed to physiological lung aging is accompanied by various host and environmental factors throughout life [55]; (b) Gene × Environment: stems from interactions between genetic (G) and environmental (E) factors across an individual’s lifespan (T) (referred to as GETomics), which can lead to lung damage or disrupt their normal developmental and aging processes [9]; (c) Comorbidity: recent findings indicate that the number of comorbidities, as assessed by the CIRS score, among patients with COPD influences the risk of moderate-to-severe exacerbations and correlates with the severity of respiratory symptoms and lung function [33]; (d) Hypoxia: Significantly contributes to COPD, increasing HIF-1α expression in hypoxic tissues and the serum inducing NF-κB expression [56]. Hypoxic environments favor the accumulation of ROS and increased oxidative stress [57]; (e) Virus infection: cigarette smoke is a well-known contributing factor to the onset and exacerbation of infectious diseases caused by viruses, such as SARS-CoV-2 [21] and influenza virus [22]; (f) Immunological changes: Current smokers have an increased inflammatory response following bacterial stimulation. The smoking effects on T cell responses persist years after individuals quit smoking, highlighting a mechanism for the persistent effects in the adaptive response associated with DNA methylation at signal trans-activators and metabolism regulators [53]; (g) Inflammation and ROS production: inflammasomes (leucine-rich repeat (LRR)-containing proteins (NLR) family members such as NLRP3, NLRP6, NLRP12, and interferon-inducible protein (AIM2)) may contribute to the development of various diseases associated with cigarette smoke exposure, including COPD. Specifically, NF-κB activation through a MyD88-dependent pathway, the production of reactive oxygen species (ROS), endoplasmic reticulum stress, mitochondrial dysfunction, and calcium influx are potentially involved in the activation of inflammasomes induced by cigarette smoke [13]. Oxidative stress is increased in COPD patients, even in former smokers and never-smokers, and is further increased during acute exacerbations; (h) DNA damage: Tobacco smoke is recognized for causing adducts on DNA, which precede DNA mutations if not repaired before DNA replication [43]. A meta-analysis conducted by Hoffmann et al. [45] suggests higher levels of DNA damage in smokers than in non-smokers; (i) Genome Instability: a systematic review and meta-analysis of studies using the micronucleus (MN) assay that measures genomic instability showed a significant association between MN frequency and the diseases investigated, and suggested a circle of events linking inflammation induced oxidative stress to the risk of disease through genomic instability and hypoxia [58].

## 4. Materials and Methods

### 4.1. Study Design and Participants

An observational cohort study was carried out in 87 patients aged 70 years or older suffering from severe COPD and admitted to the Pulmonary Rehabilitation (PR) Unit of the IRCCS San Raffaele Roma between January 2013 and December 2015 for a comprehensive 3-weeks PR program. Peripheral blood samples were collected and stored at −80 °C at admission and after 3 weeks of PR. Additional detail of the study population can be found in Russo et al. [50]. The study was approved by the ethics committee of the IRCCS San Raffaele Roma (Prot. 15/2013), and all participants signed the consent to participate in the study at admission.

All patients at admission received a European Union (EU)-validated questionnaire to estimate food items intake [59]. Given the robust evidence demonstrating beneficial effects starting from a daily consumption of vegetables [60], we compared patients eating vegetables once a day or more frequently (higher intake) with those reporting a low/moderate intake of vegetables (from 4 times a week up to a minimum of less than once a week). None answered ‘never’, and 25 patients did not answer.

All patients were stratified into three groups according to their smoking history, i.e., according to National center for health statistics (NHIS—Adult Tobacco Use—Glossary, 2019), current smoker: an adult who has smoked 100 cigarettes in his or her lifetime and who currently smokes cigarettes; former smoker: an adult who has smoked at least 100 cigarettes in his or her lifetime but who had quit smoking at the time of interview; and never-smoker: an adult who has never smoked, or who has smoked less than 100 cigarettes in his or her lifetime. These categories were confirmed after measuring the presence of cotinine in the urine samples. Consequently, we assigned subjects based on the positivity of the test rather than solely relying on their self-declarations. Urine specimens were taken at the admission into plastic containers that were subsequently frozen at −20 °C. In the urine samples, levels of cotinine were analyzed using Instant Cotinine Testing Kit by Sure Screen (Eazzi, Edinburgh, UK), certified to ISO 9001 and ISO 13485, and this drug test cassette was CE Marked in accordance with 98/79/EC [61]

### 4.2. Alkaline Comet Assay

The complete detailed procedure for the assay can be found in our previous work [62]. In summary, following new guidelines for minimal required detailed information on reporting comet assay procedures and new technical information [63], damage of DNA from lymphocytes was evaluated after lymphocytes lysation, DNA denaturation, electrophoresis on agarose gel, and staining, using the Comet assay IV software version 4 (Instem, London, UK). Tail intensity values (TI, % DNA in comet tail) were calculated from 100 comets counted for each individual.

### 4.3. Markers of Oxidative Stress

The following markers of oxidative stress were measured to evaluate the difference between smokers, current and former.

#### 4.3.1. Malonaldehyde (MDA) Assay

MDA quantification was determined as previously described by [63].

#### 4.3.2. 8-Hydroxy-2′-deoxyguanosine (8OHdG) Assay

8-hydroxy-2′-deoxyguanosine (8OHdG) was determined as previously described by [63].

### 4.4. Markers of Inflammation

Interleukin 6 (IL-6) levels and C reactive protein were determined as previously described by [63].

### 4.5. Blood Test

The number and percentage of peripheral blood cells were evaluated during routine laboratory analysis using standard blood count automated methods (Beckman Coulter LH500). LMR, NLR PLR, and SII were calculated for each patient using the absolute cell numbers of the indicated subpopulation, taken from complete blood count. Subsequently, the mean and standard deviation for observed values were calculated. An increase in NLR, PLR, and SII as well as a decrease in LMR are indicative of an ongoing inflammation [64,65,66], calculated through the following formula: “SII = Platelet × Neutrophil/Lymphocyte [39].

### 4.6. Virus Detection

#### 4.6.1. JCPyV and BKPyV

JCPyV and BKPyV DNAs were extracted from urine using DNeasy Blood and Tissue Kit (Qiagen, Milan, Italy). Extraction products were analyzed by a SYBER Green-based real-time polymerase chain reaction able to detect the JCPyV T antigen region and VP1 gene, respectively. Relative quantitative evaluation was performed by the comparative ΔΔCt method. JCPyV- and BKPyV-DNA-positive samples were further analyzed using nested-PCR for NCCR regions’ amplification. PCR products were analyzed on 2% agarose gels. The amplified products were purified using a MinElute PCR Purification Kit (Qiagen, Milan, Italy) and sequenced in a dedicated facility (Bio-Fab Research, Roma, Italy). Obtained sequences were compared with the JCPyV and BKPyV reference strains. Sequence alignment was performed using ClustalW2 on the European Molecular Biology Laboratory–European Bioinformatics Institute (EMBL-EBI) website using default parameters.

#### 4.6.2. Influenza A Virus Subtype H1N1

Total RNA from influenza A virus (H1N1) was extracted and measured with a NanoDrop spectrophotometer. Reverse-transcription (RT) and quantitative PCR (qPCR) were performed using a SensiFAST cDNA Synthesis kit (Bioline, Memphis, TN, USA) for viral HA and M2 genes. Actin was used as a reference gene for normalization. Relative quantitative evaluation was performed by the comparative ΔΔCt method.

#### 4.6.3. TTV DNA Detection and Quantification

TTV load was assessed in PMNLs where it was reported that the viral load is highest [67]. Viral DNA was extracted from blood PMNLs using the Wizard^®^Genomic DNA Purification Kit (Promega Italia S.R.L., Milan, Italy), as specified by the manufacturer. Presence and load of TTV DNA were determined in a single stepTaqMan PCR assay as described elsewhere [68]. This assay uses primers designed on a highly conserved segment of the untranslated region of the viral genome and therefore has the capacity to detect all the species in which TTV is actually classified. TTV loads were expressed as the number of viral DNA copies per μg of genomic DNA extracted by PMNLs. The lower limit of detection was 10 copies of TTVDNA per μg genomic DNA. The procedures used to quantify the copy numbers and assess specificity, sensitivity, intra- and inter-assay precision, and reproducibility have been previously described [68]. In a limited group of patients, viral DNA was also extracted from 200 μL plasma samples using a QIAamp DNA minikit^®^ (Qiagen, Chatsworth, CA, USA) and amplified as described above. The lower limit of sensitivity was 10 TTV genomes per ml of plasma.

### 4.7. Statistical Methods

Descriptive statistics of epidemiological, clinical, and biological data were represented as frequencies with percentage, mean with standard deviation for categorical and continuous variables, respectively. Group differences in demographic, clinical, and laboratory data were assessed using the Student’s *t*-test or the Mann–Whitney U-test, depending upon the distribution of variables. A statistically significant value was considered as *p* ≤ 0.05. All statistical analyses were done using the statistical software STATA 10 and SPSS (version 26.0). Survival curves were generated using GraphPad Prism 10.1.2 (GraphPad Software, Inc., San Diego, CA, USA).

## 5. Conclusions

The complexity and heterogeneity of COPD pathogenesis is confirmed by the results of the present study. In addition, this study showed a clear interaction between events which are relevant to the oxidative pathway and cigarette smoking. A category of particular interest is represented by former smokers, who in many cases experience higher damages than current smokers since subjects with more health problems are those most likely forced to quit smoking. Thus, in a similar study [69] former smokers also showed significantly greater airway obstruction than current smokers. Serim IgE concentrations were higher among smokers, but eosinophils were not significantly different between smokers and former smokers. It is possible that after smoking cessation, an inflammatory state of some type continues to degrade pulmonary function, whether cessation was due to greater health-related issues. Moreover, it is possible that after humans stop smoking, some particulate clearance may occur; the known smoking-induced suppression of macrophage M1 phenotype activation may also be relieved and permit the return to proinflammatory M1 activation, thus promoting continued or increased inflammatory and oxidative damage to the lungs among former smokers and accelerating fibrotic damage as observed in mice exposed to MnO_2_ or silica [70].

Our findings raise also the attention to other parameters which are significantly affected by smoking and can be useful to monitor in COPD patients starting a program of pulmonary rehabilitation, i.e., DNA damage, parameters of inflammation, and selected viral infections.

## Figures and Tables

**Figure 1 ijms-25-05834-f001:**
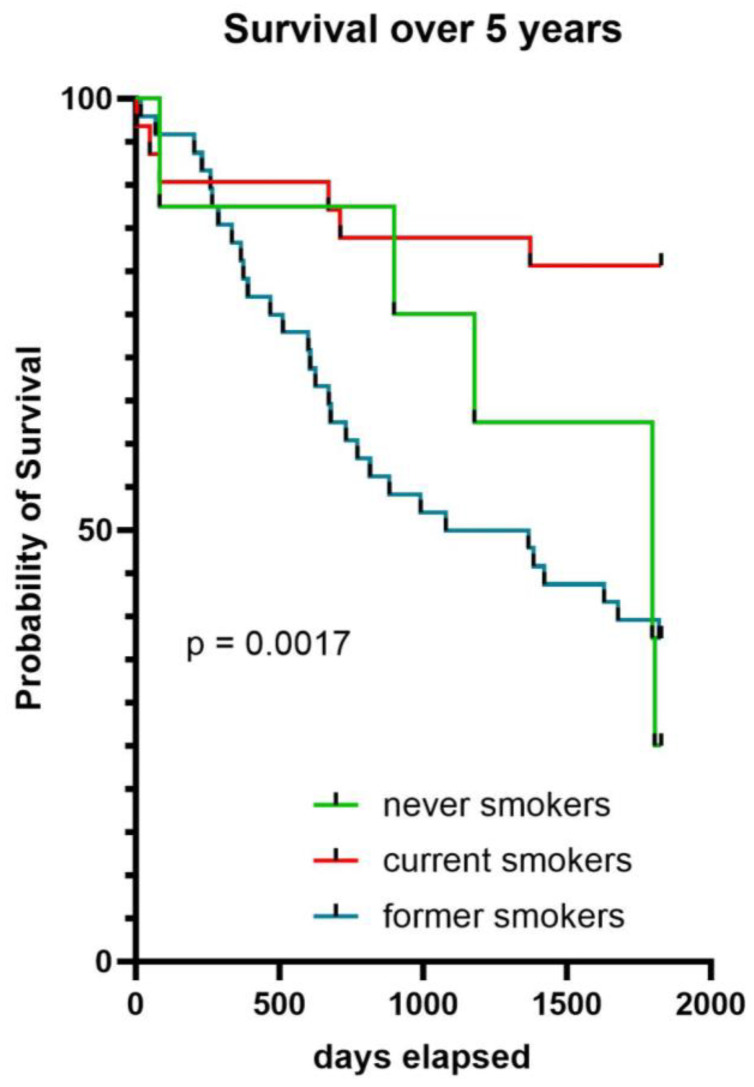
Probability of survival over 5 years in all COPD patients by smoking status.

**Figure 2 ijms-25-05834-f002:**
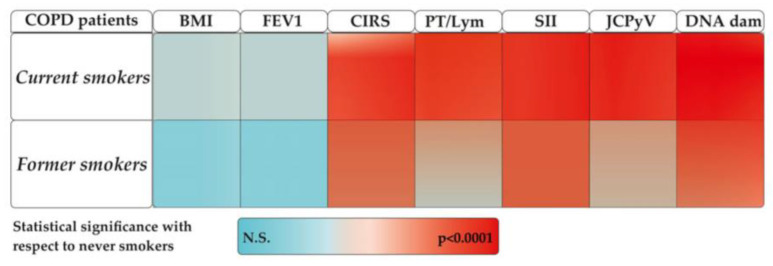
Schematic representation of some clinical and molecular parameters obtained in this study in current or former smokers compared to the same parameters obtained in non-smokers. N.S.: not statistically significant.

**Figure 3 ijms-25-05834-f003:**
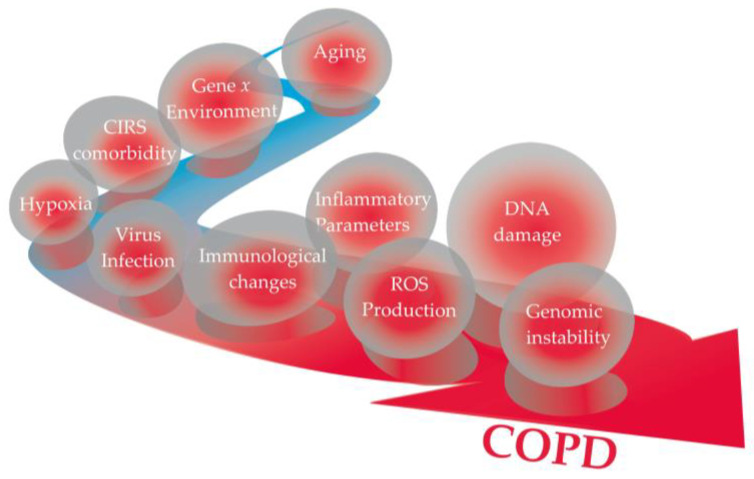
Clinical and molecular parameters involved in the etiology and progression of COPD in current smokers.

**Table 1 ijms-25-05834-t001:** COPD patients’ characteristics by smoking habit.

Variables	Total n = 87	Never- Smokers n = 10 (11.5%)	Current Smokers n = 32 (36.78%)	Former Smokers n = 45 (51.72%)	*p*-Value
Sex					NS
*Male*	40 (44.90)	5 (50)	13 (40.62)	23 (51.11)	
*Female*	47 (55.10)	5 (50)	19 (59.37)	22 (48.88)	
Age *(years)*	72.38 ± 8.74	75.50 ± 8.78	71.86 ± 9.15	71.82 ± 8.36	NS
Education					NS
*≤8 years*	52 (59.77)	7 (70.0)	10 (31.25)	30 (66.66)	
*>9 years*	29 (33.30)	--	5 (15.62)	15 (33.34)	
BMI	27.33 ± 7.33	25.17 ± 6.59	26.55 ± 4.65	28.77 ± 9.32	NS
**Clinical Features**					
Six minutes walking test (6 MWT, *meter*)	87.47 ± 87.46	82.71 ± 85.60	63.75 ± 74.25	101.00 ± 92.75	NS
Barthel Index	70.34 ± 21.77	64.63 ± 23.05	68.46 ± 24.45	72.23 ± 20.36	NS
Borg Scale Dyspnea	7.87 ± 0.92	8.25 ± 1.03	7.87 ± 0.95	7.81 ± 0.89	NS
Forced Expiratory Volume in 1 s (FEV)	48.57 ± 24.87	42.23 ± 26.12	51.69 ± 23.85	48.58 ± 25.75	NS
St. George’s Respiratory Questionnaire (SGRQ)	48.68 ± 15.90	57.80 ± 20.02	49.30 ± 15.09	47.36 ± 15.82	NS
Maugeri Respiratory Failure (MRF-26)	72.70 ± 14.65	72.21 ± 9.82	75.71 ± 15.13	71.15 ± 15.12	NS
CIRS comorbidity	2.86 ± 1.35	1.71 ± 0.76	3.58 ± 1.02	2.65 ± 1.39	0.0009
Type 2 Diabetes (N.%)	28 (32.18)	3 (30.00)	7 (21.90)	18 (40.00)	NS
Cardiovascular diseases (N. %)	38 (46.34)	4 (25.00)	16 (45.00)	18 (40.00)	NS
Oxygen Supplementation					NS
*Yes*	29 (33.30)	4 (40)	10 (31.25)	15 (33.30)	
*No*	58 (66.70)	6 (60)	22 (68.75)	30 (66.70)	
**Inflammatory Parameters**					
Lymphocytes/Monocytes	2.77 ± 198	3.28 ± 1.20	2.55 ± 2.19	2.79 ± 2.03	NS
Neutrophils/Lymphocytes	5.24 ± 3.43	4.04 ± 2.05	5.95 ± 4.05	5.10 ± 3.27	NS
Platelets/Lymphocytes	203.40 ± 152.9	132.20 ± 58.21	289.60 ± 200.2	166.10 ± 106.5	0.0009
Systemic Immune-inflammation Index (SII)	1511 ± 849.80	549.30 ± 196.70	1678 ± 816.40	1350 ± 1133	0.0258
PCR	5.89 ± 16.19	0.67 ± 0.06	0.77 ± 0.87	9.83 ± 21.00	NS
**Oxidative parameters**					
MDA (µM)	42.44 ± 13.05	39.08 ± 11.32	42.39 ± 14.00	42.96 ± 13.30	NS
8-Oxo-dG (pg/mL)	25.33 ± 11.15	20.69 ± 4.07	28.16 ± 15.69	25.18 ± 10.35	NS
IL6 (pg/mL)	72.94 ± 113.7	67.86 ± 78.76	53.57 ± 136.00	82.70 ± 107.00	NS
**DNA damage (% of Tail intensity)**					
At admission	19.47 ± 7.37	13.50 ± 3.07	25.14 ± 7.31	16.31 ± 4.48	<0.001
After 3 weeks	21.79 ± 7.26	15.02 ± 3.55	25.84 ± 5.30	20.35 ± 7.57	<0.001
**COPD patients positive for viruses**					
H1N1	1 (1.10%)	--	1 (2.80%)	--	
JCPyV	41/79 (51.90%)	8/11 (72.70%)	21/32 (65.60%)	12/36 (33.30%)	0.010
BKPyV	27/78 (34.60%)	5/11 (45.50%)	7/32 (21.90%)	15/35 (42.90%)	NS
TTV	51/87 (58.6%)	6/8 (75.0%)	21/33 (63.3%)	24/48 (50.06%)	0.05

The table reports all parameters evaluated in relation to smoking habits in COPD patients. Variables include demographic, clinical, and biochemical data in addition to DNA damage and virus infection. The table highlights differences and trends associated with smoking status. Legend: malondialdehyde (MDA), 8-Oxo-2′-deoxyguanosine (8-Oxo-dG) or interleukine-6 (IL6), influenza A virus subtype H1N1, JC virus (JCPyV), BK polyomavirus (BKPyV), Toqueteno virus (TTV), and NS: not statistically significant.

## Data Availability

The data that support the findings of this study are openly available in Zenodo at https://doi.org/10.5281/zenodo.11047757.

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
