# Peer review of "Effect of Cigarette Smoking on Clinical and Molecular Endpoints in COPD Patients"

_ijms, 2024, doi:10.3390/ijms25115834_

Round 1

Reviewer 1 Report

Comments and Suggestions for Authors

This manuscript represents a statistical analysis of a small clinical study on the physical and pathological consequences associated with COPD including molecular data related to immunological and inflammatory statuses of patients in the study. The manuscript is well written. I have only minor comments requiring attention.

1. Line 40: Systemic Immune inflammation should replace "SII".

2. Line 67: type 2 "diabetes".

3. Line 87: Low and Middle Income Countries should replace "LMIC".

4. Regarding conclusions of the authors in lines 45-47 and 483-485: In a similar size study, Goel et al. (N. Goel, B. P. Singh, N. Arora and R. Kumar, Indian J. Chest Dis. Allied Sci., 2008, 50, 329–333) former smokers also showed significantly greater airway obstruction than current smokers. Serim IgE concentrations were higher among smokers, but eosinophils were not significantly different between smokers and former smokers. It is possible that after smoking cessation, an inflammatory state of some type continues to degrade pulmonary function, whether cessation were due to greater health related issues as suggested by the authors in the abstract conclusions or due to an unbalanced selection in the final conclusions section. In fact, the finding of decreased survival, greater airways obstruction, greater pulmonary damage among former smokers than among present smokers in multiple studies suggests that the finding is not due to an unbalanced selection.

Misson et al. (P. Misson, S. van den Bruˆ le, V. Barbarin, D. Lison and F. Huaux, J. Leukocyte Biol., 2004, 76, 926–932.) observed immediate inflammatory response after acute tracheal instillation of suspended MnO2 or silica in mice. An ‘‘alternative (M2) activation’’ of murine macrophages presented in the early stages of fibrosis but returned to classical M1 activation with time and as the fibrosis progressed. Thus as particle clearance from the lungs progressed, M2 activated macrophages that suppress M1 and T cell Th1 activation decreased or reverted to M1 proinflammatory activation. It is possible that after humans stop smoking, and some particulate clearance has occurred, the known smoking induced suppression of M1 phenotype activation may be relieved also and permit the return to proinflammatory M1 activation, thus promoting continued or increased inflammatory and oxidative damage to the lungs among former smokers, thus accelerating the fibrotic damage.

Comments on the Quality of English Language

In general, English is fine. There were only a few relatively unimportant cases where wording should have been different.

Author Response

 Roma, May 22, 2024

Dr. Jaelyn Dong

Section Managing Editor, MDPI

SPECIAL ISSUE " State-of-the-Art Molecular Mechanisms of Pulmonary Pathology"

Paper: Effect of Cigarette Smoking on Clinical and Molecular End-points in COPD Patients n.  ijms-3002944

Dear Editor,

Thank you for considering our manuscript for publication. My colleagues and I found the remarks raised by the reviewers to be helpful and, we made the changes accordingly. Our response to all of the reviewers’ comments is attached.

We respectfully submit a revised version of the manuscript for publication in International Journal of Molecular Sciences SPECIAL ISSUE " State-of-the-Art Molecular Mechanisms of Pulmonary Pathology"

Yours sincerely,

On the behalf of all authors

Patrizia Russo, PhD

Unit of Clinical and Molecular Epidemiology

IRCCS San Raffaele Roma

Department of Human Sciences and Quality of Life Promotion San Raffaele University, Rome, Italy

Via di Val Cannuta, 247

00166 Rome, Italy

Tel. +39-06-52253409

Fax +39-06-52255668

 [email protected]

Reviewer comments:    

Reviewer #1: This manuscript represents a statistical analysis of a small clinical study on the physical and pathological consequences associated with COPD including molecular data related to immunological and inflammatory statuses of patients in the study. The manuscript is well written. I have only minor comments requiring attention.

  1. Line 40: Systemic Immune inflammation should replace "SII". The text was changes as requested
  2. Line 67: type 2 "diabetes". The text was changes as requested
  3. Line 87: Low and Middle Income Countries should replace "LMIC". The text was changes as requested
  4. Regarding conclusions of the authors in lines 45-47 and 483-485: In a similar size study, Goel et al. (N. Goel, B. P. Singh, N. Arora and R. Kumar, Indian J. Chest Dis. Allied Sci., 2008, 50, 329–333) former smokers also showed significantly greater airway obstruction than current smokers. Serim IgE concentrations were higher among smokers, but eosinophils were not significantly different between smokers and former smokers. It is possible that after smoking cessation, an inflammatory state of some type continues to degrade pulmonary function, whether cessation were due to greater health related issues as suggested by the authors in the abstract conclusions or due to an unbalanced selection in the final conclusions section. In fact, the finding of decreased survival, greater airways obstruction, greater pulmonary damage among former smokers than among present smokers in multiple studies suggests that the finding is not due to an unbalanced selection.

We removed the term “an unbalanced selection” and we added the sentence suggested by the reviewer: In a similar study [Goel et al. (N. Goel, B. P. Singh, N. Arora and R. Kumar, Indian J. Chest Dis. Allied Sci., 2008, 50, 329–333] former smokers also showed significantly greater airway obstruction than current smokers. Serim IgE concentrations were higher among smokers, but eosinophils were not significantly different between smokers and former smokers. It is possible that after smoking cessation, an inflammatory state of some type continues to degrade pulmonary function, whether cessation was due to greater health related issues

Misson et al. (P. Misson, S. van den Bruˆ le, V. Barbarin, D. Lison and F. Huaux, J. Leukocyte Biol., 2004, 76, 926–932.) observed immediate inflammatory response after acute tracheal instillation of suspended MnO2 or silica in mice. An ‘‘alternative (M2) activation’’ of murine macrophages presented in the early stages of fibrosis but returned to classical M1 activation with time and as the fibrosis progressed. Thus as particle clearance from the lungs progressed, M2 activated macrophages that suppress M1 and T cell Th1 activation decreased or reverted to M1 proinflammatory activation. It is possible that after humans stop smoking, and some particulate clearance has occurred, the known smoking induced suppression of M1 phenotype activation may be relieved also and permit the return to proinflammatory M1 activation, thus promoting continued or increased inflammatory and oxidative damage to the lungs among former smokers, thus accelerating the fibrotic damage.

In the revised manuscript in the conclusion we added the sentence Moreover it is possible that after humans quit smoking, particulate clearance may occur. The known smoking induced suppression of macrophage M1 phenotype activation may be relieved and permit the return to proinflammatory M1 activation, thus promoting continued or increased inflammatory and oxidative damage to the lungs among former smokers. These events may accelerate the fibrotic damage as observed in mice exposed to MnO2 or silica [P. Misson, S. van den Bruˆ le, V. Barbarin, D. Lison and F. Huaux, J. Leukocyte Biol., 2004, 76, 926–932].

Reviewer #2: ijms-3002944

The present study is beyond doubts of interest for the readers of IJMS. The results are properly presented and they are discussed clearly. However, the table 1 should be reviewed. The table contains too much data and is difficult to read them. I would recommend extracting some of the data and presenting it as a box plot for instance. The legends should be improved, I suggest to the authors to add a short description so as to catch the main message the figure wants to spread. As far as the rest is concerned, I have no further comments the study is worthy to be published

Actually, Table 1 is really crowded with information, and this can make reading difficult. On the other hand, this table provides a comprehensive view of patient’s characteristics, a piece of information quite useful to introduce our work. In the end we realized that some info was useless, since all patients received the same treatment, and therefore we removed the section on medications.

Reviewer 2 Report

Comments and Suggestions for Authors

The present study is beyond doubts of interest for the readers of IJMS. The results are properly presented and they are discussed clearly. However, the table 1 should be reviewed. The table contains too much data and is difficult to read them. I would recommend extracting some of the data and presenting it as a box plot for instance. The legends shuld be improved, I suggest to the authors to add a short description so as to catch the main message the figure wants to spread. As far as the rest is concerned, I have no further comments the study is worthy to be published

Author Response

(The authors gave the same response as above.)
